# Comparative Analysis of Hydrophilic Ingredients in Toad Skin and Toad Venom Using the UHPLC-HR-MS/MS and UPLC-QqQ-MS/MS Methods Together with the Anti-Inflammatory Evaluation of Indolealkylamines

**DOI:** 10.3390/molecules24010086

**Published:** 2018-12-27

**Authors:** Yu Zhang, Bo Yuan, Norio Takagi, Hongjie Wang, Yanyan Zhou, Nan Si, Jian Yang, Xiaolu Wei, Haiyu Zhao, Baolin Bian

**Affiliations:** 1Institute of Chinese Materia Medica, China Academy of Chinese Medical Sciences, Beijing 100700, China; zhangyuddnh@163.com (Y.Z.); hjwang@icmm.ac.cn (H.W.) yyzhou@icmm.ac.cn (Y.Z.); nsi@icmm.ac.cn (N.S.); jyang@icmm.ac.cn (J.Y.); xlwei@icmm.ac.cn (X.W.); 2Department of Applied Biochemistry, Tokyo University of Pharmacy & Life Sciences, 1432-1 Horinouchi, Hachioji, Tokyo 192-0392, Japan; yuanbo@toyaku.ac.jp (B.Y.); takagino@toyaku.ac.jp (N.T.)

**Keywords:** toad skin, toad venom, hydrophilic ingredients, UPLC-MS/MS, anti-inflammatory

## Abstract

Toad skin and toad venom, as two kinds of Chinese medicine, are prepared from *Bufo bufo gargarizans* Cantor and *Bufo melanostictus* Schneider. However, they display distinct properties in traditional application, and the hydrophilic ingredients are the possible distinguishing chemicals between them. In this work, 36 and 22 hydrophilic components were characterized from toad skin and venom, respectively, by UHPLC-HR-MS/MS, including amino acids, nucleosides, polypeptides, and indolealkylamines (IAAs). Among them, 15 compounds were unambiguously confirmed by comparison with standards. The CID-MS/MS fragmentation behaviors of seven indolealkylamine references were investigated to ascertain three types of structures. Subsequently, 11 high abundance contents of hydrophilic ingredients were determined from 11 batches of toad skin and 4 batches of toad venom by UPLC-QqQ-MS/MS. The quantitative results showed that the content of main IAAs in toad venom was much higher than in skin. In addition, the *N*-methyl serotonin (free IAA), bufothionine (combined IAA), and total IAAs sample were selected for anti-inflammatory evaluation in lipopolysaccharide (LPS) stimulated zebrafish embryo models. The obvious anti-inflammatory activities of IAAs were observed, especially for the free IAAs. This study illustrated IAAs were the main distinct hydrophilic components that probably lead to the difference between toad skin and toad venom in traditional applications.

## 1. Introduction

Toad skin and toad venom are two kinds of Chinese medicine used widely in clinical and Chinese patent medicine. Both of them are prepared from *Bufo bufo gargarizans* Cantor and *Bufo melanostictus* Schneider [1,2,3]. However, they display distinctly different properties. In traditional applications, toad skin extractions also termed as Huachansu has long been used for treating human cancers in combinational therapies [4,5], while toad venom is used for treating infection and inflammation as well as relieving pain and inducing resuscitation [6,7,8]. Previous study has proven minor polar chemical, bufadienolide, is a common component of these two preparations, and is also the main chemical that exerts anticancer activity. Hydrophilic ingredients are regarded as the potential distinct constituents of these two preparations [9]. However, the reason behind this difference is far from being understood.

Based on the literature, the water-soluble ingredients separated or detected from toad skin and venom contain mainly indolealkylamines (IAAs), nucleosides, steroids, and polypeptides [10,11]. Evidence has supported that IAAs are the predominant hydrophilic ingredients in toad skin. Due to the lack of standards, only several IAAs have previously been quantified by HPLC [12,13,14]. In a qualitative study, an ion trap mass spectrometer was used for analysis [15]. However, it took a much longer time and provided an inaccurate molecular weight of the constituents. Along with the development of analytical instruments, ultra high performance liquid chromatography (UHPLC) coupled with mass spectrometry makes it possible to perform rapid qualitative and quantitative analysis of natural products because of the high resolution, sensitivity, and efficiency [11,16].

As the major hydrophilic constituents from toad skin and venom [17], IAAs present a high affinity for the 5-hydroxytryptamine receptor 2A (5-HT_2A_), with antibacterial, antimalarial, antidepressant, and anxiolytic activities [18,19,20,21]. Specifically, serotonin displays its antagonistic activity by conjugating with peripheral serotonin receptors [22]. Bufotenine was used as a snuff for the psychological effect, which also showed acetylcholine-like activity affecting the firing pattern of most of the neurons tested (dose 20 µg/mL). The action of bufotenidine was similar, although less potent than that of acetylcholine [23,24]. Additionally, bufothionine had a good inhibitory effect, especially on HepG2, by inducing cell apoptosis and arresting cells at the G_2_ phase [25]. It can also induce proteins for mitochondria-mediated apoptosis, inhibit liver tumors and protect the liver against acute injury [26].

In recent years, zebrafish have become a popular in vivo model for assessing drug efficiency, toxicity, and safety, with the advantages of short life cycle, high reproductive rate, low cost, and easy care [27,28,29]. The LPS-induced zebrafish embryo has been proven to be a successful inflammatory model [30], which has been used for the evaluation of caffeine, oleuropein, and phillyrin for their anti-inflammatory responses [27,29,31].

In this paper, the hydrophilic ingredients were compared between toad skin and venom based on UHPLC-HR-MS and UPLC-QqQ-MS/MS, respectively. The anti-inflammatory effect of their main difference compounds IAAs were also evaluated with LPS triggered zebrafish larvae to illustrate the reason of their difference in traditional applications.

## 2. Results and Discussion

### 2.1. Fragmentation Behaviors of the IAAs

A total of seven IAAs standards were investigated to ascertain their CID-MS/MS fragmentation routes. They were assigned to three different types: type I linear IAAs, type II β-carboline IAAs, and type III quinolin IAAs. The fragmentation pathways of the aforementioned compounds were found to assist in the identification of unknown compounds.

#### 2.1.1. Type I Linear IAAs

Serotonin, *N*-methyl serotonin, bufotenine, and bufotendine were the representative linear IAAs. Four precursor ions [M + H]^+^ were detected at *m*/*z* 177.1021 (C_10_H_13_ON_2_, Cal. 177.1022, error 0.79 ppm), *m*/*z* 191.1181 (C_11_H_15_ON_2_, Cal. 191.1178, error 0.52 ppm), *m*/*z* 205.1337 (C_12_H_17_ON_2_, Cal. 205.1335, error 0.78 ppm), and *m*/*z* 219.1490 (C_13_H_19_ON_2_, Cal. 219.1492, error 0.87 ppm) in their full mass spectra, respectively. All four of these compounds displayed similar fragmentation patterns. After the loss of the corresponding alkyl amine, the diagnostic ion of *m*/*z* 160.0759 (C_10_H_10_ON) was obtained. Then, this obtained ion further generated the common ions at *m*/*z* 134.0603 (C_8_H_8_ON), *m*/*z* 132.0808 (C_9_H_10_N), *m*/*z* 117.0573 (C_8_H_7_N), and *m*/*z* 115.0541 (C_9_H_7_) by loss of C_2_H_2_, CO, C_2_H_3_O, and CH_3_ON residues, respectively. Meanwhile, the [M + HC_2_H_5_N]^+^ ion at *m*/*z* 148.0757 was detected in the MS/MS analysis of *N*-methyl serotonin (Figure 1). The unsaturation number of the aforementioned compounds is 6.

#### 2.1.2. Type II β-Carboline IAAs

Shepherdine was the representative β-carboline IAAs. A precursor ion [M + H]^+^ was detected at *m*/*z* 203.1176 (C_12_H_15_ON_2_, Cal. 203.1179, error 1.43 ppm) in the full mass spectrum. After losing an NH_3_ group, the product ion at *m*/*z* 186.0911 [M + H − NH_3_]^+^ was observed in the MS/MS analysis, which further dissociated to the [M + H − NH_3_ − C_2_H_2_]^+^ ion at *m*/*z* 160.0755. Meanwhile, the fragment ion [M + H − NCH_3_]^+^ at *m*/*z* 174.0911 and the [M + H − NCH_3_ − C_2_H_2_]^+^ ion at *m*/*z* 148.0754 were detected (Appendix A). The degree of unsaturation of shepherdine is 7.

#### 2.1.3. Type III Quinolin IAAs

Bufothionine was the representative quinolin IAAs. Accurate mass analysis yielded an [M + H]^+^ ion at *m*/*z* 283.0747 (C_12_H_15_O_4_N_2_S, Cal. 283.0747, error 0.01 ppm). Its fragment ion at *m*/*z* 203.1186 [M + H − SO_3_]^+^ was obtained after loss of one SO_3_. The product ion at *m*/*z* 203.1186 could be further fragmented into ions at *m/z* 188.0949 [M + H − SO_3_ − CH_3_]^+^, *m*/*z* 174.0909 [M + H − SO_3_ − NCH_3_]^+^, and *m*/*z* 160.0762 [M + H − SO_3_ − C_2_H_5_N], respectively. Dehydrobufotenine was the other quinolin IAAs which had a similar fragmentation pattern to bufothionine. The precursor ion [M + H]^+^ at *m*/*z* 203.1175 (C_12_H_14_ON_2_, Cal. 203.1178, error −1.92 ppm) produced fragment ions [M + H − CH_3_]^+^, [M + H − NCH_3_]^+^, and [M + H − C_2_H_5_N]^+^ at *m*/*z* 188.0940, *m*/*z* 174.0909, and *m*/*z* 160.0759, respectively. Specifically, dehydrobufotenine could produce another fragment ion [M + H − C_2_H_3_N]^+^ at *m*/*z* 162.0909 (Appendix A). Like the β-carboline indole alkylamine, the degree of unsaturation of the quinolin IAAs is also 7.

Generally, all the IAAs could produce the fragment ion at *m*/*z* 160.0759. The corresponding intensity of this characteristic ion was much higher in the linear IAAs than the other two types. Additionally, the diagnostic ion (C_11_H_12_ON) at *m*/*z* 174.0912 was only detected in the type II and III IAAs. In addition, the fragment ions at *m*/*z* 132.0808 (C_11_H_12_ON), *m*/*z* 117.0573 (C_8_H_7_N) and *m*/*z* 115.0541 (C_9_H_7_) were the common fragments only detected in type I. Furthermore, the unsaturation number was beneficial for the identification of the unknown IAAs.

### 2.2. Qualitative Analysis of Hydrophilic Ingredients

#### 2.2.1. Optimization of Extraction Conditions

To achieve a satisfactory separation for the hydrophilic ingredients, an ACQUITY UPLC^®^ T3 column was optimized for the advantage of 100% water phase tolerance and the avoidance of peak tailing. Three types of extraction solvents (water, 50% ethanol, 50% methanol) were evaluated to obtain high extraction efficiency. Finally, 50% methanol was selected for analysis consideration the higher response value of the interest peak and the lower interference of other peaks (Appendix A).

#### 2.2.2. Identification of Hydrophilic Ingredients

For the enrichment of the hydrophilic ingredients from the toad skin and venom extraction, the ODS column was used for preparation. A 30% methanol fraction was selected for UHPLC-HR-MS/MS analysis. A total of 38 compounds have been identified from toad skin and venom (Figure 2), among which, 15 compounds were confirmed by comparison with standards. Twenty unknown compounds were preliminarily identified by comparison of their fragmentation and retention time as reported in literature [1,11,32,33,34,35,36]. Most amino acids and nucleosides were identified by standards (Table 1). In this study, we focused on the identification of IAAs. The identification of three typical IAAs was given as follows.

Peak 19 was detected at 3.87 min with a precursor ion [M + H]^+^ at *m*/*z* 203.1171 (C_12_H_15_ON_2_, Cal. 203.1165, error −3.89 ppm), which was identified as an isomer of shepherdine. The diagnostic ion at *m*/*z* 160.0754 in the MS/MS spectrum also supported that it was an IAAs. By comparison with fragmentation of shepherdine, the characteristic ion [M + H − NH_3_]^+^ at *m*/*z* 186.0913 was not observed. However, the fragment ion at *m*/*z* 174.0907 (C_11_H_12_ON) was found by loss of the NCH_3_ residue, which indicated that the methyl was probably attached to the nitrogen atom of the side chain. Therefore, peak 19 was identified as 1-methyl-6-hydroxy-1,2,3,4-tetrahydro-β-carboline (Appendix A).

Peak 20 was observed at 4.32 min with the precursor ion [M + H]^+^ at *m*/*z* 219.1120 (C_12_H_15_O_2_N_2_, Cal. 219.1128, error −3.67 ppm). According to the diagnostic ion at *m*/*z* 160.0754 in the MS/MS analysis, we first speculated that it was an IAAs. The unsaturation number was 7, which indicated that it should contain a carbonyl group or a new hexatomic ring. The fragment ion at *m*/*z* 191.1171 (C_11_H_15_ON_2_) was yielded by the loss of one CO group. The rest of the product ions were the same as *N*-methyl serotonin. A carbonyl group was found to be attached to the nitrogen atom on the side chain of *N*-methyl serotonin. Based on the aforementioned information, peak 20 was identified as *N*-(2-(5-hydroxy-1H-indol-3-yl) ethyl)-*N*-methylformamide (Appendix A).

Peak 31 was detected at 10.46 min with the [M + H]^+^ ion at *m*/*z* 201.1023 (C_12_H_13_ON_2_, Cal. 201.1022, error −0.30 ppm), which displayed two less hydrogens than shepherdine. Its unsaturation number was 8, and a double bond should be present in the structure. In the MS/MS spectrum, the fragment ions at *m*/*z* 186.0786 (C_11_H_10_ON_2_) and *m*/*z* 160.0757 (C_10_H_10_ON) were observed after the loss of a methyl group and a C_2_H_3_N residue, respectively. Therefore, Peak 33 was identified as 1-methyl-2,9-dihydro-1H-pyrido [3–β] indol-6-ol (Appendix A).

According to the qualitative results, a total of 36 and 22 hydrophilic ingredients, including amino acids, purine, peptides, and IAAs, were preliminarily identified from toad skin and venom, respectively. Amino acids are the raw materials for the synthesis of tissue proteins and can also transform carbohydrate and fat [37]. Purine is an important base compound of nucleic acid, and the end product of metabolism in vivo is uric acid. When uric acid deposits in the body, hyperuricemia and gout are likely to occur [38]. Peptides have been reported to have antiviral and anti-tumor biological activities [39,40]. Among these compounds we detected, there were 17 and 10 IAAs were included in toad skin and venom, respectively. Which indicated that IAAs were the predominant hydrophilic compounds in these two preparations. Studies have shown that IAAs may be associated with antimicrobial activity [41]. Due to the relative content of such components were different in toad skin and venom. That may be the reason why the two preparations have been used differently.

### 2.3. Quantitative Analysis of Hydrophilic Ingredients

To further compare the hydrophilic ingredients and find the compounds with significant difference between toad skin and venom, 11 high abundance content hydrophilic ingredients, including 6 IAAs, were selected for the quality assessment, all of which were visible in the total ion chromatography.

#### 2.3.1. Optimization of Mass Spectrometric Conditions

The mass spectrometric conditions (DP: declustering potential; CE: collision energy; EP: entrance potential; CXP: collision cell exit potential) of 11 compounds were optimized (Appendix A). The optimized chromatogram is shown in Figure 3.

#### 2.3.2. Method Validation

The quantitative analysis method was validated by specificity, linearity, limit of detection (LOD), limit of quantitation (LOQ), intraday and interday precision, accuracy, and stability. Specifically, the LOD and LOQ were evaluated with signal-to-noise ratios (S/N) of 3 and 10, respectively. Moreover, the recovery test was set at low (80% of original amount), medium (100% of original amount) and high (120% of original amount) levels (*n* = 3).

Calibration curves of 11 standards showed good linearity (r from 0.9988 to 0.9999 and 0.9985 to 0.9999). The LOD and LOQ for each constituent in this experiment were 0.0069–25.88 ng/mL and 0.047–103.52 ng/mL, respectively. Both intraday and interday precision was evaluated. The RSDs of 11 quality markers were less than 3.38%. Furthermore, the sample solutions were prepared in parallel (*n* = 6) to evaluate the repeatability and achieved RSDs of 0.71–3.21% for toad skin and 0.36–2.93% for toad venom. The stability was tested by analyzing the ratio of peak area to internal standard of each sample extract at the ambient temperature for 0, 2, 4, 8, 10, 12, and 24 h. The RSDs were less than 3.46% and 3.31%, respectively (Appendix A). The recovery of the 11 constituents ranged from 96.18% to 104.56% and from 95.80% to 102.53% for the two preparations. The RSD values were from 0.09% to 3.08% and from 0.73% to 3.21%, respectively (Appendix A).

All of these results demonstrated that the UPLC-QqQ-MS/MS method was precise, stable and accurate enough, which met the requirements for the determination of 11 constituents in toad skin and venom.

#### 2.3.3. Quantitative Analysis of 11 Quality Markers

The determination results of 11 quality markers are shown in Table 2. Significant variations of 11 quality markers were observed in toad skin and toad venom. The contents of bufotenidine (0.9796–2.8443 mg/g) and bufotenine (0.6567–4.7657 mg/g) were much higher than the other constituents in all the toad skin sample. At the same time, bufotenidine (134.6315–215.3163 mg/g) was observed as the predominant compound in the toad venom samples. Overall, the total content of IAAs in toad venom (224.4381–292.9211 mg/g) was much higher than in toad skin (3.0694–7.2399 mg/g). To find the specific quality markers with significant differences between the two preparations, a multivariate statistical analysis was carried out. As shown in Figure 4, according to the PCA score plot, we found that the two preparations had a clear clustering trend. After analysis by OPLS-DA, the clustering trend became more obvious. Subsequently, the VIP plot displayed five components with significant differences (VIP > 1). They were bufotenidine, *N*-methylserotonin, bufotenine, dehydrobufotenine, and adenine. It was obvious that IAAs were the major distinct compounds between toad skin and venom.

Among the six main IAAs we detected, *N*-methyl serotonin, bufotenidine, bufotenine, dehydrobufotenine were free IAA, bufothionine was combined IAA. The main difference of these two category IAAs was if the OH attached in C-5 was replaced by sulfate radical. In general, the quantitative results give us a hint that free IAAs maybe the main chemicals lead to the difference of application of these two preparations. The activity of these constituents deserved further evaluation.

### 2.4. Anti-Inflammation Effect of IAAs in the LPS-Induced Zebrafish Model

In this experiment, three doses (25, 50, and 100 μg/mL) of total IAAs enriched from toad skin (Number TS-SC-1), *N*-methyl serotonin (free IAA) and bufothionine (combined IAA) were selected to evaluate their anti-inflammatory effect in LPS-stimulated zebrafish embryos. The content of *N*-methyl serotonin and bufothionine in total IAAs was 1.02 mg/g and 9.27 mg/g, respectively.

As shown in Figure 5A, the number of neutrophils in the yolks represents the extent of inflammation. By comparison with the control group (CG), severe neutrophil infiltration was observed in the yolks of the LPS treated group. Generally, a dose-dependent increase in activity was observed in all the test groups. The total IAAs showed obvious anti-inflammatory effects at the dose of 50 μg/mL, compared with the LPS group (*p* < 0.01). The ratio of anti-inflammatory activity was 35%, it was better than that of positive medicine (indomethacin) with the effect was 25%. When the dose was 100 μg/mL, the activity reached 55% (*p* < 0.001). Specifically, treatment with *N*-methyl serotonin displayed a significant ability to decrease the number of neutrophils at all three concentrations (25, 50, and 100 μg/mL), with the activity at 35, 55, and 60% (*p* < 0.001), respectively (Figure 5B,C). However, the bufothionine group showed a relatively weaker anti-inflammatory effect with the activity at 30% in the 100 μg/mL dosage (*p* < 0.05).

The results aforementioned indicated that the constituents of IAAs did have obvious anti-inflammatory activity, the effect of which even better than positive control group when the concentration over 50 μg/mL. In addition, free IAA *N*-methyl serotonin showed better activity than combined IAA bufothionine. Given that the content of *N*-methyl serotonin in total IAAs was much smaller than bufothionine, which led us speculate that free IAAs were the main compounds exert anti-inflammatory activity, probably. Furthermore, we have known that the total content of IAAs, especially free IAAs, in toad venom is much higher than in skin, that is maybe the reason why toad venom was used for treating infection and inflammation in traditional application. By the way, the anti-inflammatory activity of IAAs was reported for the first time.

## 3. Conclusions

In this study, an effective and comprehensive strategy was developed for the comparative evaluation of hydrophilic ingredients in toad skin and venom, with a total of 38 components characterized. IAAs were detected as the predominant constituents in both preparations. Quantitative results showed that the total content of main IAAs in toad venom was much higher than in toad skin. Importantly, free IAAs were proved the main different constituents between them. In the efficacy investigation, total IAAs showed obvious anti-inflammation activity, among which, the effect of free IAA *N*-methyl serotonin was better than combined IAA bufothionine. This text manifested that toad skin and toad venom were different in traditional application may results from the different content of IAAs, especially free IAAs. The corresponding mechanism of the anti-inflammatory activity of IAAs should be studied in depth in the future.

## 4. Material and Methods

### 4.1. Chemical and Reagents

Acetonitrile and methanol of HPLC grade were obtained from Fisher Scientific (Fair Lawn, NJ, USA); deionized water was produced by a Milli-Q water system (Millipore, Bedford, MA, USA), and MS grade formic acid was purchased from Thermo Fisher Scientific Products (Thermo Fisher Scientific, San Jose, CA, USA); other chemicals and solvents were of analytical grade. LPS, dimethyl sulfoxide (DMSO), and methyl cellulose were purchased from Sigma-Aldrich Co. (St. Louis, MO, USA)

The reference standards—including valine, adenine, nicotinic acid, hypoxanthine, and xanthine—were purchased from BaoJi Herbest Bio-Tech Co., Ltd. (ShanXi, China), and the purities of these standards were >98%. Serotonin, *N*-methyl serotonin, bufotenidine, bufotenine, dehydrobufotenine, and bufothionine were isolated from the crude extract of toad skin in the preliminary experiment. The purities of these standards were above 98% according to HPLC-UV analysis (Fair Lawn, NJ, USA).

### 4.2. Materials

Toad skin (S1–S11) and toad venom (S12–S15) were collected from the market (Sichuan, Shandong, Jiangsu, Henan, Shanxi, and Beijing Provinces, China, in March 2017). All crude materials were identified by Prof. Hongjie Wang and deposited in the Institute of Chinese Materia Medica, China Academy of Chinese Medical Sciences.

### 4.3. Preparation of Samples and Standard Solution

#### 4.3.1. Qualitative Sample Preparation

Accurately weighed toad skin powder (50 mg) and toad venom powder (10 mg) were extracted by ultrasonication for 30 min with 50% methanol at a concentration of 50 mg/mL and 10 mg/mL, respectively, to prepare the samples. The extracted solution was condensed to 100 μL, applied to an Octadecylsilyl gel column (ODS, 1 mL, 1 cm × 2 cm) and eluted with 30% methanol for 5 mL. The elution was filtered through a 0.22 μm membrane filter for UHPLC-HR-MS/MS analysis.

#### 4.3.2. Quantitative Sample Preparation

Accurately weighed toad skin powder (50 mg) and toad venom powder (10 mg) were extracted by ultrasonication for 30 min with 50% methanol at a concentration of 50 mg/mL and 10 mg/mL, respectively, to prepare the samples. After centrifugation for 10 min at 12,000 rpm, the supernatants were filtered through a 0.22 μm membrane filter for UPLC-QqQ-MS/MS analysis.

#### 4.3.3. Anti-Inflammatory Sample Preparation

A total of 100 g of toad skin powder was extracted by ultrasonication in 1000 mL of 50% methanol for 30 min. The extracting solution was condensed to 5 mL, and applied to an Octadecylsilyl gel column (ODS, 150 mL, 8.0 × 16 cm) eluted with water and 30% methanol. All 750 mL of the elution was collected as one fraction. The 30% methanol fraction was employed total IAAs for anti-inflammatory activity analysis.

#### 4.3.4. Standard Solution Preparation

Standard stock solutions of valine, adenine, nicotinic acid, hypoxanthine, xanthine, serotonin, *N*-methyl serotonin, bufotenidine, bufotenine, dehydrobufotenine, and bufothionine were accurately weighted and dissolved in 50% methanol at a concentration of 1 mg/mL. The solutions were stored at 4 °C until needed for analysis.

### 4.4. Fish Maintenance and Care

Transgenic neutrophilic fluorescent zebrafish (10 and 12 months) of both sexes were purchased from Hunter Biotechnology, Inc. (HangZhou, China). They were maintained at 28 °C on a 14:10 h (day/night) light cycle and fed three times daily. Embryos (3 dph) can be obtained from individual fish by pairwise breeding in a male-to-female ratio of 4:1. Before the experiment started, the embryos were anesthetized using tricaine methanesulfonic acid (Sigma Aldrich, Steinheim, Germany), followed by an injection of 10 mg/mL LPS into the yolk with a volume of 10 nL for each embryo to establish the inflammation model. Indomethacin (Indo) (Shanghai Jingchun Industrial Co., Ltd., Shanghai, China) was used as a positive control (28.6 µg/mL). After washing by embryo media three times, LPS-stimulated zebrafish larvae were treated with the total IAAs, *N*-methyl serotonin (free IAAs) and bufothionine (bound IAAs) solution at three concentrations (25, 50, and 100 µg/mL) for 3 h at 28 °C. Finally, ten zebrafish larvae randomly selected from each group were put in a 3% methylcellulose solution. Images were obtained by an electrofocusing zoom microscope (AZ100, Nikon, Tokyo, Japan) and analyzed using Nikon NIS-Elements D 3.10 Advanced Image Processing Software (NIS-Elements D 3.10, Nikon, Tokyo, Japan) to calculate the neutrophils number (N). The experiments met the requirements of international AAALAC certification.

The formula for calculating the anti-inflammatory rate of the test product is as follows: Anti-inflammatory rate (%) = (N (model group) − N (test group))/N (model group) × 100%. (N is the number of neutrophil).

Statistical processing results are expressed as the mean ± standard error (SE). Statistical analysis was performed using GraphPad prism 5 (GraphPad Software, Inc. Version 5.01, La Jolla, CA, USA). A one-way analysis of variance (ANOVA) with Dunnett’s *t*-test was carried out for statistical comparison, with *p* < 0.05 indicating a significant difference.

### 4.5. UHPLC-HR-MS/MS and UPLC-QqQ-MS/MS Conditions

#### 4.5.1. UHPLC-HR-MS/MS Analysis

An ultimate 3000 hyperbaric liquid chromatography (LC) system coupled with a LTQ Orbitrap mass spectrometer (Thermo Fisher Scientific, San Jose, CA, USA) was used in this study. An UPLC reverse-phase C18 analytical column (2.1 × 100 mm, 1.8 μm, ACQUITY UPLC^®^ T3, Waters, Milford, MA, USA) was used to analyze the samples. The mobile phase consisted of A (water containing 0.1% formic acid) and B (acetonitrile). The linear gradient conditions were optimized as follows: 0–6.0 min, 3% B; 6.0–8.0 min, 3–10% B; 8.0–9.0 min, 10–20% B; 9.0–22.0 min, 20–50% B; and 22.0–30.0 min, 50–95% B. The flow rate was 0.3 mL/min, and the column temperature was maintained at 35 °C. The injection volume was 1 μL. Mass detection was performed in positive ion mode. The ESI source parameters were as follows: ion spray voltage, +5.0 kV; sheath gas flow rate, 35 arb; aux gas flow rate, 10 arb; capillary temperature, 350 °C; S-lens RF level, 60%. The mass resolution of Fourier transform (FT) was 30,000 with a full scan in the range of *m*/*z* 50–1000. The MS/MS and MS3 experiments were set as data-dependent scans. Data processing was performed using Xcalibur 3.0 (Thermo Fisher Scientific, San Jose, CA, USA), Metworks 1.3 (Thermo Fisher Scientific, San Jose, CA, USA) and Mass Frontier 7.0 software packages (Thermo Fisher Scientific, San Jose, CA, USA).

#### 4.5.2. UPLC-QqQ-MS/MS Conditions

A Waters ACQUITY UPLC^®^ system (Waters, Milford, MA, USA) coupled with an AB SCIEX QTRAP^®^ 6500 (AB SCIEX, California, CA, USA) mass spectrometer with ESI in multiple reaction monitoring (MRM) detection mode was used for quantitative analysis of the toad skin and venom samples. A Waters ACQUITY HSS T3 column (2.1 × 100 mm, 1.8 μm, ACQUITY UPLC^®^ T3, Waters, USA) was performed with the column temperature set at 35 °C. The mobile phase consisted of A (water containing 0.1% formic acid) and B (acetonitrile). The linear gradient conditions were optimized as follows: 0–2.5 min, 0.1–0.5% B; 2.5–2.6 min, 0.5–3% B; 2.6–7.0 min, 3–5% B; 7.0–8.0 min, 5–95% B; and 8.0–11.0 min, 95–95% B. The flow rate was 0.4 mL/min, and the injection volume was 1 μL. A QTRAP^®^ 6500 MS equipped with an electrospray ionization (ESI) source (AB SCIEX, Los Angeles, CA, USA) was performed in positive ion mode with a source temperature of 550 °C. The curtain gas (CUR) was set to 35 psi. The parameters of MS and MS/MS analysis—including declustering potential (DP), collision energy (CE), and collision cell exit potential (CXP)—were optimized.

#### 4.5.3. Statistical Analysis

Quantitative analysis was performed by Analyst^®^ software (version 1.6.2, Berkeley, CA, USA) and MultiQuantTM software (version 3.0, San Francisco, CA, USA). The results were expressed as the mean value ± standard deviation of triplicate data. The resulting data were exported to SIMCA software (MKS UMETRICS AB, version 14.1.0, Umetrics, Umeå, Sweden) for PCA, partial least squared discriminant analysis (PLS-DA), and VIP analysis.

## Figures and Tables

**Figure 1 molecules-24-00086-f001:**
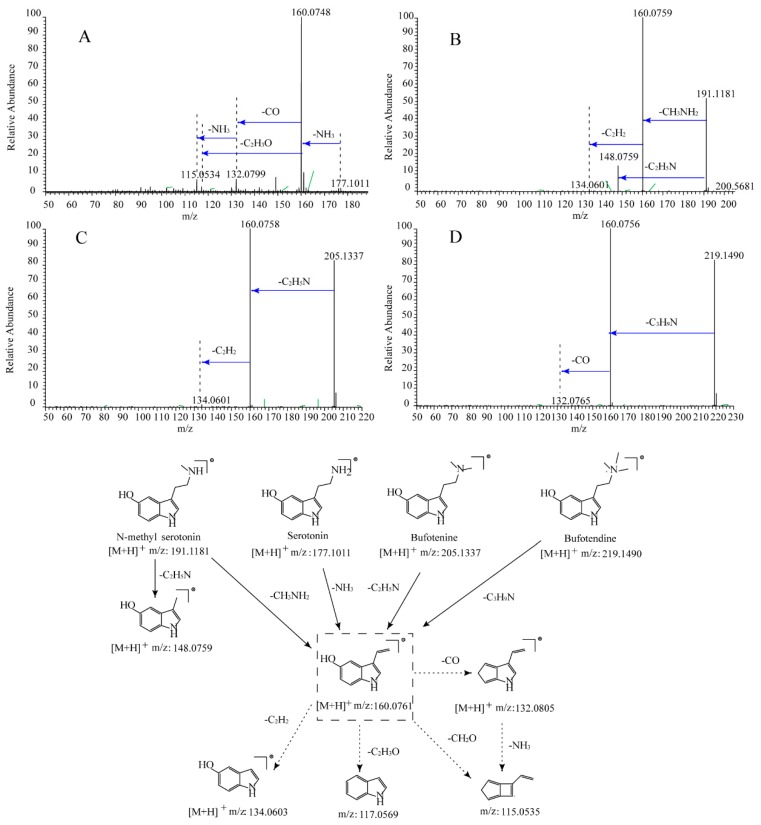
MS/MS spectrum and fragmentation patterns of (**A**): Serotonin; (**B**): *N*-methyl serotonin; (**C**): Bufotenine; and (**D**): Bufotenidine.

**Figure 2 molecules-24-00086-f002:**
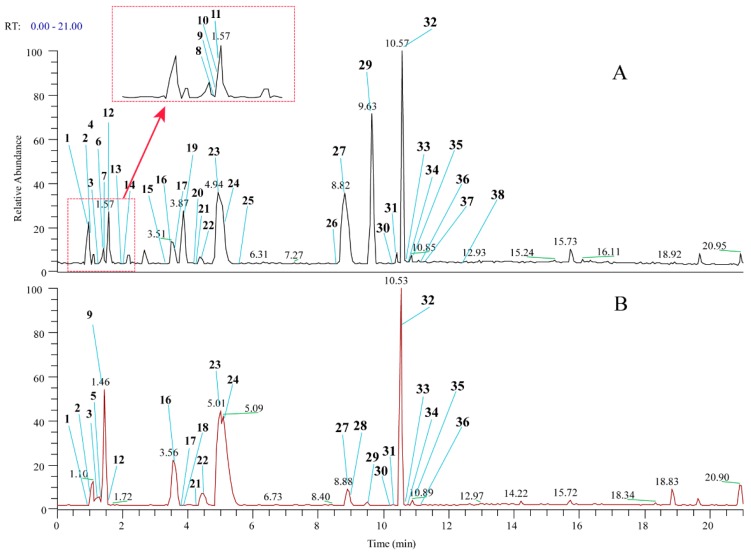
Total ion chromatograms (TIC) of 30% methanol elution fraction in the positive-ion mode (from UHPLC-MS) of (**A**): toad skin; (**B)**: toad venom. The peak numbering in each TIC relates to numbered compounds listed in Table 1.

**Figure 3 molecules-24-00086-f003:**
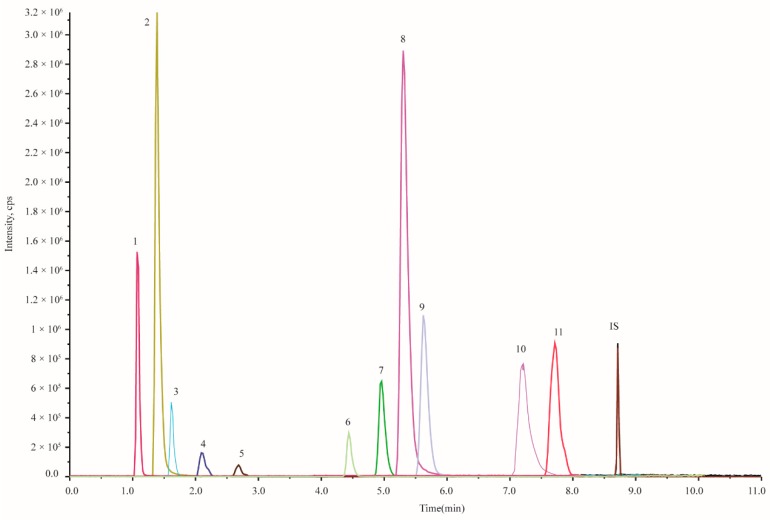
The MRM chromatogram of mixed standards (11 markers and 1 internal standard). 1. Valine; 2. Adenine; 3. Nicotinic acid; 4. Hypoxanthine; 5. Xanthine; 6. Serotonin; 7. *N*-methyl serotonin; 8. Bufotenidine; 9. Bufotenine; 10. Dehydrobufotenine; 11. Bufothionine; IS. Diazepam.

**Figure 4 molecules-24-00086-f004:**
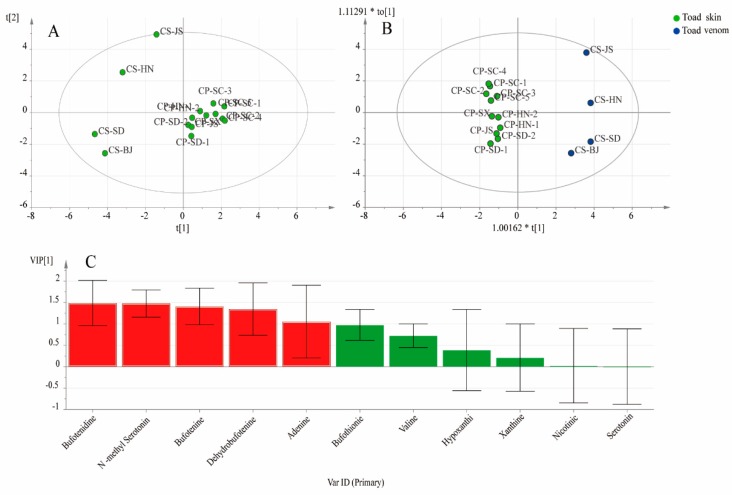
Multivariate statistical analysis of toad skin and toad venom samples. (**A**): PCA score plot; (**B**): OPLS-DA score plot; (**C**): VIP plot.

**Figure 5 molecules-24-00086-f005:**
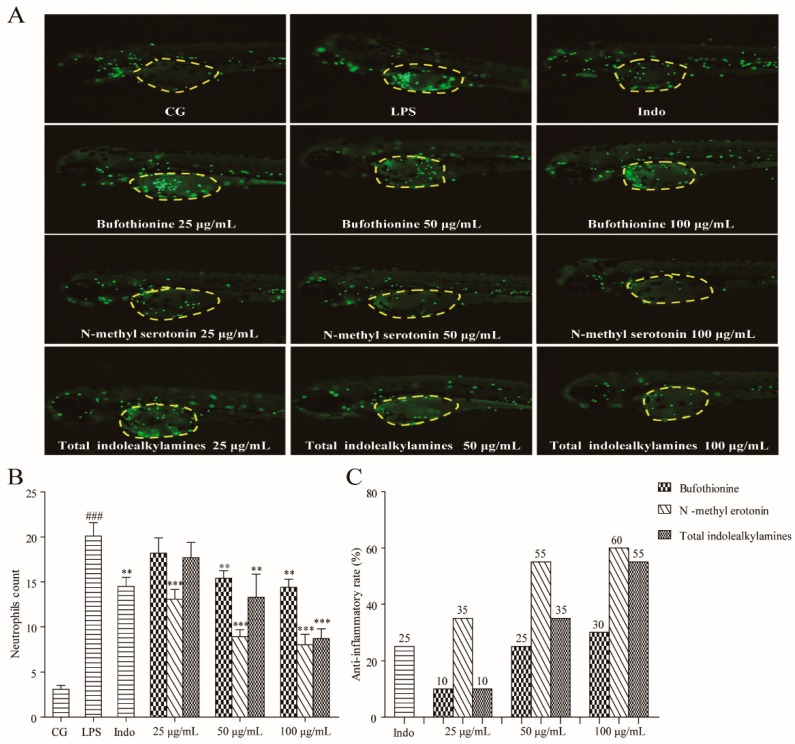
Bufothionine, *N*-methyl serotonin, and total indolealkylamines inhibit neutrophil recruitment induced by LPS. (**A**): Fluorescence imaging of 3-dpf zebrafish live larvae (*n* = 10). The green dots in the yellow circles indicate the neutrophil. The control group (CG), LPS and indomethacin group (Indo) (28.6 µg/mL) are shown. (**B**): Number of neutrophils. (**C**): Effect of anti-inflammatory in each group. * *p* < 0.05, ** *p* < 0.01 and *** *p* < 0.001 compared with the LPS group.

**Table 1 molecules-24-00086-t001:** Hydrophilic ingredients identified in toad skin and toad venom extracts by UHPLC-HR-MS/MS.

Peak	T_R_ (min)	Selective Ion	Full Scan MS (*m*/*z*)		Mass Error (ppm)	Formula	MS/MS Data	Identification	Toad Skin	Toad Venom
			Measured	Theory						
1	0.97	[M + H]^+^	118.0859	118.0862	−2.33	C_5_H_11_O_2_N	101.0593, 83.0486	* valine	+++	+
2	1.00	[M + H]^+^	132.10149	132.1019	−3.14	C_6_H_13_O_2_N	90.0545	leucine	+	+
3	1.30	[M + H]^+^	136.0613	136.0617	−2.81	C_5_H_5_N_5_	119.0346, 94.0398	* adenine	++	+
4	1.34	[M + H]^+^	123.0554	123.0552	0.9	C_6_H_6_ON_2_	105.0330, 82.0135	nicotinamide	+	-
5	1.35	[M + H]^+^	178.0866	178.0862	1.99	C_10_H_11_O_2_N	160.0759, 132.0803	5-Hydroxy tryptophol	-	++
6	1.38	[M + H]^+^	254.0890	254.0897	2.64	C_9_H_11_O_4_N_5_	236.0784, 218.0677, 206.06770, 194.0676,	eritadenine	++	-
7	1.38	[M + H]^+^	124.0389	124.0393	−2.54	C_6_H_5_O_2_N	96.0437, 80.0487	* nicotinic acid	++	-
8	1.40	[M + H]^+^	152.0570	152.0566	2.19	C_5_H_5_ON_5_	135.0302	guanine	++	-
9	1.46	[M + H]^+^	275.1345	275.1350	−1.73	C_10_H_18_O_5_N_4_	240.0985, 215.1031, 175.1186	succinyl arginine	+	+++
10	1.47	[M + H]^+^	155.0810	155.0815	−2.99	C_7_H_10_O_2_N_2_	127.0866, 113.96368	cyclo(pro-gly)dipeptide	+	-
11	1.49	[M + H]^+^	113.0342	113.0345	−2.69	C_4_H_4_O_2_N_2_	-	* uracil	+	-
12	1.57	[M + H]^+^	137.0456	137.0457	−0.78	C_5_H_4_ON_4_	119.0158	* hypoxanthine	+++	+
13	1.84	[M + H]^+^	153.0410	153.0407	2.27	C_5_H_4_O_2_N_4_	136.0143, 115.5171, 110.0348	* xanthine	++	-
14	2.51	[M + H]^+^	127.0504	127.0502	1.7	C_5_H_6_O_2_N_2_	109.0280	thymine	+	-
15	3.24	[M + H]^+^	189.1015	189.1009	−3.44	C_11_H_12_ON_2_	172.0758, 160.07594, 146.0601, 132.0772	* 1,2,3,4-tetrahydro-6-hydroxy-β-carboline	++	-
16	3.51	[M + H]^+^	177.1021	177.1022	−0.79	C_10_H_12_ON_2_	160.07541, 132.0799, 117.0569, 115.0534	* serotonin	+++	+++
17	3.61	[M + H]^+^	303.1667	303.1663	1.5	C_12_H_22_O_5_N_4_	285.1561, 268.1295, 250.1189, 225.1236, 175.1187	adipyl arginine	++	++
18	3.77	[M + H]^+^	221.0912	221.0920	−3.7	C_11_H_12_O_3_N_2_	−204.0648, 186.0542, 162.0544	5-hydroxy tryptophan	-	+
19	3.87	[M + H]^+^	203.1171	203.1165	−3.89	C_12_H_14_ON_2_	174.0907, 160.0747	^#^ 2-methyl-6-hydroxy-1,2,3,4-tetrahydro-β-carboline	++	-
20	4.32	[M + H]^+^	219.1122	219.1128	−3.67	C_12_H_14_O_2_N_2_	191.1171, 160.0752, 148.0752, 132.0803	^#^*N*-(2-(5-hydroxy-1H-indol-3-yl)ethyl)-*N*-methylformamide	+	-
21	4.34	[M + H]^+^	205.0966	205.0971	−2.46	C_11_H_12_O_2_N_2_	176.0695, 160.0754	*N*′-formylserotonin	+	+
22	4.38	[M + H]^+^	191.1181	191.1178	−0.52	C_11_H_14_ON_2_	160.0759, 148.0759, 134.0601	* *N*-methyl serotonin	+++	+++
23	4.94	[M + H]^+^	219.1490	219.1492	0.87	C_13_H_18_ON_2_	160.0756, 132.0765	* bufotenidine	+++	+++
24	5.11	[M + H]^+^	205.1337	205.1335	−0.78	C_12_H_16_ON_2_	160.0758, 134.0601	* bufotenine	+++	++
25	5.58	[M + H]^+^	203.1176	203.1179	1.43	C_12_H_14_ON_2_	186.0911, 174.0911,160.0755, 148.0754	* shepherdine	+	-
26	8.56	[M + H]^+^	191.1174	191.1178	−2.46	C_11_H_14_ON_2_	160.0754	*O*-methyl serotonin	+	-
27	8.82	[M + H]^+^	203.1174	203.1178	−1.92	C_12_H1_4_ON_2_	188.0940, 174.0908, 162.0909, 160.0761,	* dehydrobufotenine	+++	+
28	8.87	[M + H]^+^	317.1830	317.1830	−0.86	C_13_H_24_O_5_N_4_	300.1562, 282.1456, 264.1348, 239.1395, 175.1186	pimeloyl arginin	+++	++
29	9.63	[M + H]^+^	283.0741	283.0740	0.01	C_12_H_14_O_4_N_2_S	203.1186, 188.0949, 174.0909, 160.0762	* bufothionine	+++	+
30	10.35	[M + H]^+^	203.1167	203.1178	0.54	C_12_H_14_ON_2_	188.0888, 174.0914, 160.0759	cyclotufotenine	+	-
31	10.46	[M + H]^+^	201.1023	201.1022	−0.30	C_12_H_12_ON_2_	186.0786, 160.0757	^#^ 1-methyl-2,9-dihydro-1H-pyrido [3–b]indol-6-ol	+	+
32	10.57	[M + H]^+^	331.1971	331.1976	−1.38	C_14_H2_6_O_5_N_4_	278.1492, 250.1545, 175.1185	suberoyl arginine	+++	+++
33	10.72	[M + H]^+^	192.0650	192.0655	−2.55	C_10_H_9_O_3_N	160.0753, 113.9634	(5-Hydroxy-1H-indol-3-yl)acetic acid	+	+
34	10.77	[M + H]^+^	345.2146	345.2154	0.12	C_15_H_28_O_5_N_4_	327.2040, 303.1924, 282.1821,267.1711, 228.1602, 189.1350,175.1186	azelayl arginine	++	++
35	10.84	[M + H]^+^	359.2277	359.2275	0.55	C_16_H_31_O_5_N_4_	341.2178, 328.1843, 289.1753, 271.1647, 253.1542.172.1076	sebacyl arginine	++	++
36	11.15	[M + H]^+^	195.0870	195.0876	−3.09	C_8_H_11_O_2_N_4_	138.0657, 110.0707	* caffeine	+	+
37	11.20	[M + H]^+^	219.1122	219.1128	−2.48	C_12_H_14_O_2_N_2_	160.0754	* 5-methoxy bufotenine	++	-
38	12.43	[M + H]^+^	433.2188	433.2193	−1.35	C_20_H_29_O_5_N_6_	160.0746	bufobutarginine	+	-

* Structures confirmed by comparison with reference standards. ^#^ First detected in toad skin or toad venom; +++, high content; ++, medium content; +, low content; -, not detected. Bold characters: the base peaks in MS/MS spectra.

**Table 2 molecules-24-00086-t002:** Contents of the 11 hydrophilic ingredients in toad skin and toad venom samples (μg/g, *n* = 3).

No.	Sample Code	Valine	Adenine	Nicotinic Acid	Hypoxanthine	Xanthine	Serotonin	*N*-methyl Serotonin	Bufotenidine	Bufotenine	Dehydrobufotenine	Bufothionine
S1	TS-SC-1	289.28 ± 1.53	24.93 ± 0.34	63.81 ± 0.86	351.63 ± 0.84	213.34 ± 1.17	247.86 ± 1.68	202.38 ± 2.32	2023.47 ± 8.30	898.80 ± 3.98	418.66 ± 2.11	1062.02 ± 24.37
S2	TS-SC-2	254.24 ± 3.39	8.94 ± 0.06	59.86 ± 0.79	299.48 ± 1.39	133.16 ± 1.57	217.47 ± 1.62	304.57 ± 5.27	2844.32 ± 28.11	2231.71 ± 20.29	496.79 ± 3.62	925.18 ± 4.43
S3	TS-SC-3	205.23 ± 3.94	15.41 ± 0.28	66.98 ± 1.76	356.57 ± 2.13	202.68 ± 0.82	262.52 ± 7.36	106.30 ± 2.67	1323.30 ± 6.80	732.22 ± 9.43	325.26 ± 4.61	610.68 ± 7.56
S4	TS-SC-4	713.48 ± 1.88	2.58 ± 0.06	45.22 ± 0.50	299.62 ± 2.60	253.07 ± 1.18	170.37 ± 2.14	186.58 ± 3.16	1574.04 ± 11.72	1566.19 ± 30.63	498.23 ± 0.95	637.58 ± 6.82
S5	TS-SC-5	402.55 ± 1.31	24.53 ± 0.82	62.00 ± 1.33	299.47 ± 1.89	145.66 ± 0.62	244.31 ± 1.78	136.32 ± 1.67	1144.70 ± 35.00	904.18 ± 3.43	384.24 ± 1.06	695.51 ± 1.82
S6	TS-SD-1	33.03 ± 0.33	3.37 ± 0.05	50.01 ± 0.51	131.71 ± 1.44	35.27 ± 0.91	189.82 ± 1.79	293.10 ± 4.22	1156.39 ± 9.70	1231.26 ± 33.43	218.06 ± 1.26	348.16 ± 5.82
S7	TS-SD-2	64.81 ± 0.96	52.50 ± 1.35	59.91 ± 1.09	154.76 ± 0.40	30.2 ± 0.72	232.60 ± 4.76	468.57 ± 4.68	1479.88 ± 16.68	1881.36 ± 11.32	115.98 ± 0.61	263.31 ± 10.29
S8	TS-HN-1	43.89 ± 0.40	55.98 ± 0.28	60.62 ± 1.13	285.93 ± 1.78	89.91 ± 1.59	234.12 ± 4.71	351.01 ± 5.42	1676.05 ± 24.61	1350.93 ± 37.80	102.65 ± 1.03	305.47 ± 8.21
S9	TS-HN-2	179.89 ± 4.45	32.74 ± 1.01	69.00 ± 1.01	305.86 ± 2.91	90.54 ± 1.10	271.23 ± 4.38	327.04 ± 3.03	1293.41 ± 9.34	1877.24 ± 17.11	125.13 ± 2.25	374.96 ± 5.75
S10	TS-JS	71.64 ± 1.18	48.33 ± 0.69	56.29 ± 0.22	201.72 ± 1.33	58.91 ± 0.76	216.93 ± 0.84	685.63 ± 4.24	979.57 ± 6.02	4765.70 ± 30.99	150.75 ± 0.58	441.37 ± 6.50
S11	TS-SX	138.97 ± 0.58	16.89 ± 0.26	68.68 ± 0.76	210.95 ± 0.54	114.54 ± 2.24	270.65 ± 3.37	70.54 ± 1.27	1063.14 ± 26.91	656.70 ± 7.80	418.95 ± 8.19	589.47 ± 9.07
S12	TV-SD	23.68 ± 0.29	407.13 ± 1.17	23.16 ± 0.77	216.52 ± 2.08	91.29 ± 1.06	85.55 ± 1.33	14,337.25 ± 19.33	134,631.52 ± 2932.45	51,542.82 ± 189.25	23,643.68 ± 332.84	197.23 ± 3.80
S13	TV-HN	24.70 ± 0.74	225.69 ± 4.04	96.90 ± 1.61	393.48 ± 2.08	54.06 ± 1.44	374.67 ± 1.02	8529.53 ± 30.17	194,375.84 ± 2139.65	24,517.40 ± 109.79	48,021.88 ± 650.30	251.46 ± 6.88
S14	TV-JS	11.48 ± 0.24	20.54 ± 0.41	108.85 ± 1.46	762.27 ± 3.70	242.87 ± 2.67	448.66 ± 6.13	11,570.16 ± 243.42	204,441.63 ± 736.45	25,478.04 ± 266.90	16,349.94 ± 150.75	125.72 ± 0.50
S15	TV-BJ	18.91 ± 0.28	114.16 ± 0.86	8.41 ± 0.14	56.90 ± 1.16	10.35 ± 0.12	22.38 ± 0.41	11,422.40 ± 68.57	215,316.39 ± 4687.76	43,488.38 ± 416.28	22,589.69 ± 776.03	81.83 ± 1.64

Abbreviations: Toad skin (TS); Toad venom (TV).

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
