# Peer review of "Comparative Analysis of Hydrophilic Ingredients in Toad Skin and Toad Venom Using the UHPLC-HR-MS/MS and UPLC-QqQ-MS/MS Methods Together with the Anti-Inflammatory Evaluation of Indolealkylamines"

_molecules, 2018, doi:10.3390/molecules24010086_

Round 1

Reviewer 1 Report

In the present study, the authors comparatively investigated the chemical difference and the anti-inflammatory effects of two Chinese medicines of toad skin and toad venom. The results are interesting and very helpful for the further development and clinical application of toad skin and toad venom, as well as their related products. The design of the experimental is reasonable and the manuscript is well organized. I suggest that this manuscript could be accepted for publication in the journal of MOLECULES after a major revision.

1. The results of the study should be presented in the section of RESULTS and DISCUSSION, but not in the section of INTRODUCTION. Therefore, the last paragraph in the section of INTRODUCTION (page 2) should be moved into the section of RESULTS and DISCUSSION.

2. To compare the anti-inflammatory effects of IAAs in the LPS-induced zebrafish model, total IAAs enriched from toad skin, N-methyl serotonin (free IAAs) and bufothionine (bound IAAs) were selected with the same concentrations (25, 50, and 100 μg/mL). However, the contents of N-methyl serotonin and bufothionine in total IAAs enriched from toad skin should be detected, and the contributions of the methyl serotonin and bufothionine to total IAAs on anti-inflammation should be further compared and discussed. Moreover, the difference on anti-inflammation between toad skin IAAs and toad venom IAAs should be further compared and discussed.

3. Toad skin and toad venom are two different Chinese Medicines and with different applications in clinics. The pharmacological activities are considered to be associated to their chemical profiles. In this study, the chemical profiles of toad skin and toad venom were comparatively investigated both qualitatively and quantitatively. Moreover, the difference in chemistry and therapeutic applications of toad skin and toad venom should be further discussed in the manuscript based on the detected results in chemistry.

Reviewer 2 Report

«Comparative analysis of hydrophilic ingredients in toad skin and toad venom using the UPLC-HR- MS/MS and UPLC-QqQ-MS/MS methods together with the anti-inflammatory evaluation of indolealkylamines» by Zhang et al.

The manuscript describes the identification and quantification of components in two different toad preparations (venom and skin) using HR-MS/MS and QqQ-MS/MS, respectively. Selected compound were also tested for anti-inflammatory properties using a zebrafish assay.

The manuscript is in general well written, and English spelling and style is mostly fine. However, there are some spelling errors that should be corrected.

Specific comments:

Abstract

- Use «high abundance compounds» instead of «high level compounds»

- Use «lay» instead of «laid»

Introduction

- Put references 9-11 in one bracket

- UPLC is a trademark for Waters. UHPLC should be used when discussing technology in general

- Use «selected for quantitation» instead of «selected for determination»

- Avoid use of the term «excellent»

Section 2.2.1

- The title indicate that the UHPLC-separation was optimized, but the text gives no details. Instead it is mentioned that different extraction solvents were tested. Please make a clear distinction between optimizing chromatographic, extraction and mass spectrometric parameters.

- Why was 50% metanol chosen? Please explain.

- What does «50% methanol» means? Aquatic methanol?

- Concerning optimization of the gradient: Why does the gradient run for 30 min when the last compound of interest elutes after approx. 12 min?

Figure 2.

- Provide information in the figure text on what kind of analysis the figure shows

Figure 3

- What were the criteria for the classification of abundance? This should be specified in the text.

- Replace «didn’t detected» with «not detected»

Section 2.3.1

- Again, please distinguish between optimizing chromatographic and mass spectrometric condition.

- Explain what DP, CE and CXP refers to

Section 2.3.2

- Consider using the term «two preparations» instead of «two medicines»

Section 2.4

- Why was not the two different preparations tested for AIF activity? If the purpose of the study was to understand the differences between the two preparations (skin and venom), it is strange that their activities were not compared.

- What was used as source for the total IAAs? This should be specified in the text (not only in materials&methods)

- Why were the two compounds selected for AIF testing? This should be rationalized. 

- What is shown in fig 5C? I do not understand the use of the term «anti-inflammatory activity». This must be explained, and the title of the y-axis of fig 5C must be specified.

Section 4.2

- What does «all crude materials were identified..» mean? What is the details of this process?

Conclusions

- The authors should try to discuss some possible connections between the chemical analysis they have done and the different activities of the preparations (the traditional use of venom and skin preparations). Do they believe that the compounds they have found to be present in different amounts is responsible for the different activities? It is a pity that the two different preparations are not tested for AIF-activities.

Figure S3

- It is three solvents, not four as indicated in the text

Formatting issues

- All formulae in the text should have numbers in subscript

- m/z should be in italics

- The «+» in the adducts should be in superscript

- In many places a space should be inserted between the last letter of a sentence and the reference

Author Response

December. 21th, 2018

Dear reviewer,

Thank you for your consideration of our manuscript entitled “Comparative analysis of hydrophilic ingredients in toad skin and toad venom using the UHPLC-HR-MS/MS and UPLC-QqQ-MS/MS methods together with the anti-inflammatory evaluation of indolealkylamines”. We are pleased that you gave positive comments and acknowledged the scientific merit of our work.

We have carefully studied the comments and revised the text substantially following these comments. We believe the concerns have been addressed (see Point-by-point response to the reviewer’s comments attached).

Again, thanks for your attention and your help in facilitating the reviewing process.

Sincerely yours,

Haiyu Zhao

Institute of Chinese Materia Medica, China Academy of Chinese Medical Sciences, Beijing, 100700, China

Point-by-point response to the reviewer’s comments

Response to reviewer 2

Thank you for your good suggestions to improve our manuscript. We have re-checked and revised the potential grammatical mistakes and minor errors in the text. The detailed revisions were listed as follows.

Abstract

- Use «high abundance compounds» instead of «high level compounds»

Response: Thank you for your suggestion. we have revised “high abundance compounds” instead of “high level compounds”.

- Use «lay» instead of «laid»

Response:

Thank you for your suggestion. The last sentence has been deleted in the revised manuscript.

Introduction

- Put references 9-11 in one bracket

Response: Thank you for your suggestion. we have put references 9-11 in one bracket.

- UPLC is a trademark for Waters. UHPLC should be used when discussing technology in general

Response: Thank you for your suggestion. I agree with you, and have changed UPLC with UHPLC in the revised manuscript.

- Use «selected for quantitation» instead of «selected for determination»

Response: Thank you for your suggestion. We have changed selected for determination with selected for quantitation.

- Avoid use of the term «excellent»

Response: I agree with your suggestion, We have changed excellent with obvious.

Section 2.2.1

- The title indicate that the UHPLC-separation was optimized, but the text gives no details. Instead it is mentioned that different extraction solvents were tested. Please make a clear distinction between optimizing chromatographic, extraction and mass spectrometric parameters.

Response: Thank you for your suggestion. We have changed the title “Optimization of chromatographic conditions” with “Optimization of extraction conditions” in 2.2.1.

- Why was 50% methanol chosen? Please explain.

Response: Thank you for your question. As is shown in figure S3, we compared the total ion flow chromatography with consideration of the higher response value of the interest peak and lower interference of other peaks. It was found that the best effect was obtained by 50% methanol extraction. Therefore, 50% methanol was selected for analysis.

Figure S3. Optimized extraction conditions selected from three extraction solvents. A: water; B: 50% methanol; C: 50% ethanol.

- What does «50% methanol» means? Aquatic methanol?

Response: 50% methanol means the volume of water and methanol is 1:1.

- Concerning optimization of the gradient: Why does the gradient run for 30 min when the last compound of interest elutes after approx. 12 min?

Response: Thank you for your question. When we optimized the extract conditions, the sample we used was extracted by 50% methanol, gradient run for 30 min was the final condition. However, in order to better analyze the composition of hydrophilic constituents (peaks from 0 min to 13 min), we enriched the fraction eluted by 30% methanol from toad skin and venom (Figure 2). The condition of gradient was the same one (30 min) (Figure S3).

Figure S3. Optimized extraction conditions selected from three extraction solvents. A: water; B: 50% methanol; C: 50% ethanol.

Figure 2. The 30% methanol fraction from toad skin and toad venom. A: toad skin; B: toad venom.

Figure 2.

- Provide information in the figure text on what kind of analysis the figure shows

Response: Thank you for your suggestion. We have provide information as “Total Ion Chromatograms (TIC) of 30% methanol elution fraction in the positive-ion mode (from UHPLC- MS) of (A): toad skin; (B): toad venom. The peak numbering in each TIC relates to numbered compounds listed in Table 1.” in the figure text.

Figure 3

- What were the criteria for the classification of abundance? This should be specified in the text.

Response: Thank you for your suggestion. High abundance compounds in this paper means the main peaks which were visible in the total ion chromatography. We have explained and supplemented in 2.3.

- Replace «didn’t detected» with «not detected»

Response: Thank you for your comment. We have replaced “didn’t detected” with “not detected”.

Section 2.3.1

- Again, please distinguish between optimizing chromatographic and mass spectrometric condition.

Response: Thank you for your comment. We have changed the title “Optimization of chromatographic parameters and sample preparation” with “Optimization of mass spectrometric conditions”.

- Explain what DP, CE and CXP refers to

Response: Thank you for your suggestion. We have supplement the full name of abbreviation with “DP: declustering potential; CE: collision energy; EP: entrance potential; CXP: collision cell exit potential”.

Section 2.3.2

- Consider using the term «two preparations» instead of «two medicines»

Response: Thank you for your suggestion. We have revised “two preparations” instead of “two medicines”.

Section 2.4

- Why was not the two different preparations tested for AIF activity? If the purpose of the study was to understand the differences between the two preparations (skin and venom), it is strange that their activities were not compared.

Response: Thank you for your suggestion. First, the production of toad venom is low and the price is expensive, so it’s hard to enrich the total IAAs from this product. In this text, The purpose of enrichment of total IAAs was to verify the anti-inflammatory activity of these substances. Qualitative and quantitative results showed that the structure type of IAAs was similar in this two preparations, only the content was different. Two IAAs (N-methyl serotonin and bufothionine) with different contents in toad skin and venom were selected for the AIF activity evaluation. The results proved that the effect of free IAA N-methyl serotonin was better than that of combined IAA bufothionine. Specifically, the content of N-methyl serotonin in toad skin (70.54-685.63 μg/g) was much lower than in venom (8529.53-14337.25 μg/g), while, bufothionine was higher in toad skin (263.31-1062.02 μg/g) than in toad venom (81.83-251.46 μg/g). Based on the evidence aforementioned, we speculate that the AIF activity of total IAAs from toad venom might be better.

- What was used as source for the total IAAs? This should be specified in the text (not only in materials&methods)

Response: Thank you for your suggestion. We have supplement the source of total IAAs in the text. “The total IAAs was enriched from the toad skin (TS-SC-1)

- Why were the two compounds selected for AIF testing? This should be rationalized.

Response: Thank you for your good question. The reason why we choose N-methyl serotonin and bufothionine for AIF testing is that the 6 main IAAs we detected in toad skin and venom can be divided into two categories, combined IAAs (represented by bufothionine) and free IAAs (represented by N-methyl serotonin). The main difference of their structure is that if the OH of C-5 was replaced by sulfate radical. On the other hand, the content of N-methyl serotonin in toad skin was much lower than in venom, while, bufothionine was higher in toad skin than in toad venom. Two compounds with different content in this two preparations may be more representative.

- What is shown in fig 5C? I do not understand the use of the term «anti-inflammatory activity». This must be explained, and the title of the y-axis of fig 5C must be specified.

Response: Thank you for your good question. We have revised the title of the y-axis of fig 5C “anti-inflammatory activity” as “anti-inflammatory rate”. There is a formula to calculate the anti-inflammatory rate: Anti-inflammatory rate (%)=(N(model group) -N (test group))/N(model group) *100%. N is the number of neutrophil.

Section 4.2

- What does «all crude materials were identified..» mean? What is the details of this process?

Response: Thank you for your question. “all crude materials were identified..” means that all the sample material of toad skin and venom in this paper were identified by our professor before we start our study. The identification method was based on the morphological identification of traditional Chinese medicine, mainly consideration of the color, shape, smell of toad skin, the density of warts on the back, the size of the gland behind the ear and so on to identify the materials was genuine.

 Conclusions

- The authors should try to discuss some possible connections between the chemical analysis they have done and the different activities of the preparations (the traditional use of venom and skin preparations). Do they believe that the compounds they have found to be present in different amounts is responsible for the different activities? It is a pity that the two different preparations are not tested for AIF-activities.

Response: Thank you for your suggestion. In accordance with your opinion, we have supplemented the discussion about the connections between the chemical and activities of the preparations in the Results and discussion section of each experiment. The reason why we didn’t compare the AIF-activities has been illustrated in previous question.

Figure S3

- It is three solvents, not four as indicated in the text

Response: Thank you for your suggestion. We have corrected this error.

Formatting issues

- All formulae in the text should have numbers in subscript

Response: Thank you for your suggestion. We have modified the formula of the whole paper according to your opinion.

- m/z should be in italics

Response: Thank you for your good suggestion. We have modified the font format of “m/z” in the whole paper according to your opinion.

- The «+» in the adducts should be in superscript

Response: Thank you for your good comment. We have modified the “+” in the whole paper according to your opinion.

- In many places a space should be inserted between the last letter of a sentence and the reference

Response: Thank you for your comments. We have rechecked and modified the space for the whole text according to your opinion.

Reviewer 3 Report

In this study was developed a comparative evaluation of hydrophilic ingredients in toad skin and venom, with a total of 38 components characterized. IAAs were detected as the predominant constituents. A validated quality control method was established using UPLC-QqQMS/MS. A multivariate statistical analysis was carried out to find the specific compounds with significant differences between toad skin and venom. Also, the authors show that N-methyl serotonin, showed excellent anti-inflammatory activity.

          The manuscript fits within the scope of the journal. The manuscript is interesting and the idea is very nice. The author’s work on discussing achieved results is appreciated. Some revisions are necessary to improve clarity of the paper.

         I have some recommendations for authors:

- The purpose of the paper is not clearly highlighted in the abstract or in the text of the manuscript (introduction). The authors refer more to the results obtained, without clearly delineating of the aim/hypothesis of the paper.

-  Because it's interesting for journal readers, I kindly ask the authors to complete the introduction with information on current uses of toad skin and toad venon. There are numerous bibliographies that refer to its biological action. For exemple, skin extraction have a real potential as resources for preventing or treating human cancers by inducing apoptosis, sensitizing cancer cells to conventional cancer therapies, or protecting host cells from any side effects. Please see: doi: [10.4103/0973-1296.137358], doi.org/10.1016/j.jep.2016.03.062,doi:10.30638/eemj.2018.088,http://dx.doi.org/10.1155/ 2014/312684,  etc.

 - If is possible, please include some information about possible mechanisms of IAAs in cells. Some studies have shown that toad medicines decrease inflammation through a variety of mechanisms, including inhibition of NF- ?B and its signalling molecules and pathways.

 -  Include in the text potential research directions.

Author Response

December. 21th, 2018

Dear reviewer,

Thank you for your consideration of our manuscript entitled “Comparative analysis of hydrophilic ingredients in toad skin and toad venom using the UHPLC-HR-MS/MS and UPLC-QqQ-MS/MS methods together with the anti-inflammatory evaluation of indolealkylamines”. We are pleased that you gave positive comments and acknowledged the scientific merit of our work.

We have carefully studied the comments and revised the text substantially following these comments. We believe the concerns have been addressed (see Point-by-point response to the reviewer’s comments attached).

Again, thanks for your attention and your help in facilitating the reviewing process.

Sincerely yours,

Haiyu Zhao

Institute of Chinese Materia Medica, China Academy of Chinese Medical Sciences, Beijing, 100700, China

Point-by-point response to the reviewer’s comments

Response to reviewer 3

Thank you for your good suggestions to improve our manuscript. We have revised the first paragraph by requoting some of the literature, and made the purpose clear in this text. The detailed revisions were listed as follows.

- The purpose of the paper is not clearly highlighted in the abstract or in the text of the manuscript (introduction). The authors refer more to the results obtained, without clearly delineating of the aim/hypothesis of the paper.

Response: Thank you for your suggestion. We have modified the first paragraph of Introduction as follows.

“Toad skin and toad venom are two kinds of Chinese medicine used widely in clinical and Chinese patent medicine. Both of them are prepared from Bufo bufo gargarizans Cantor and Bufo melanostictus Schneider [1-3]. However, they display distinctly different properties. In traditional applications, toad skin extractions also termed as Huachansu has long been used for treating human cancers in combinational therapies [4, 5], while toad venom is used for treating infection and inflammation as well as relieving pain and inducing resuscitation [6-8]. Previous study have proved minor polar chemicals bufadienolide are common components of this two preparations, which are also the main chemicals that exerts anticancer activity. Hydrophilic ingredients are regarded as the potential distinct constituents of this two preparations [9]. However, the reason lead to this difference is far from being understood.”

So, our hypothesis was that hydrophilic ingredients were the potential distinct constituents lead to their different in application. The purpose of this text is to clarify the reason why toad skin and toad venom were different in traditional application. At last, we proved our hypothesis based on qualitative and quantitative study combined with anti-inflammatory activity evaluation experiment.

- Because it's interesting for journal readers, I kindly ask the authors to complete the introduction with information on current uses of toad skin and toad venom. There are numerous bibliographies that refer to its biological action. For example, skin extraction have a real potential as resources for preventing or treating human cancers by inducing apoptosis, sensitizing cancer cells to conventional cancer therapies, or protecting host cells from any side effects. Please see: doi: [10.4103/0973-1296.137358], doi.org/10.1016/j.jep.2016.03.062,doi:10.30638/eemj.2018.088,http://dx.doi.org/10.1155/ 2014/312684,  etc.

Response: Thank you for your good suggestion. The biological actions have been illustrated in the revised manuscript. The details were as follows.

“Toad skin and toad venom are two kinds of Chinese medicine used widely in clinical and Chinese patent medicine. Both of them are prepared from Bufo bufo gargarizans Cantor and Bufo melanostictus Schneider [1-3]. However, they display distinctly different properties. In traditional applications, toad skin extractions also termed as Huachansu has long been used for treating human cancers in combinational therapies [4, 5], while toad venom is used for treating infection and inflammation as well as relieving pain and inducing resuscitation [6-8]. Previous study have proved minor polar chemicals bufadienolide are common components of this two preparations, which are also the main chemicals that exerts anticancer activity. Hydrophilic ingredients are regarded as the potential distinct constituents of this two preparations [9]. However, the reason lead to this difference is far from being understood.”

- If is possible, please include some information about possible mechanisms of IAAs in cells. Some studies have shown that toad medicines decrease inflammation through a variety of mechanisms, including inhibition of NF- ?B and its signalling molecules and pathways.

- Include in the text potential research directions.

Response: Thank you for your suggestion. Actually, your opinion is also what we plan to do next. We've already started this part of the experiment, while, it’s still need some time to finish this work.

Reviewer 4 Report

Molecules 410321 Comparative analysis of hydrophilic ingredients in toad skin and toad venom using the UPLC-HRMS/MS and UPLC-QqQ-MS/MS methods together with the anti-inflammatory evaluation of indolealkylamines. The manuscript is excellent in all its development. The work makes a good contribution to analytical chemistry. The methodology is adequate and the results are solid and supported by experimental work. I would only suggest an extension in the evaluation section of anti-inflammatory activity. Authors should include the data of the positive control used in the experiment (recognized anti-inflammatory compound), and include in the discussion.

Author Response

December. 21th, 2018

Dear reviewer,

Thank you for your consideration of our manuscript entitled “Comparative analysis of hydrophilic ingredients in toad skin and toad venom using the UHPLC-HR-MS/MS and UPLC-QqQ-MS/MS methods together with the anti-inflammatory evaluation of indolealkylamines”. We are pleased that you gave positive comments and acknowledged the scientific merit of our work.

We have carefully studied the comments and revised the text substantially following these comments. We believe the concerns have been addressed (see Point-by-point response to the reviewer’s comments attached).

Again, thanks for your attention and your help in facilitating the reviewing process.

Sincerely yours,

Haiyu Zhao

Institute of Chinese Materia Medica, China Academy of Chinese Medical Sciences, Beijing, 100700, China

Point-by-point response to the reviewer’s comments

Response to reviewer 4

Thank you for your good suggestions to improve our manuscript. We have made corresponding modifications to the problems you mentioned in the revised manuscript. The detailed revisions were listed as follows.

Molecules 410321 Comparative analysis of hydrophilic ingredients in toad skin and toad venom using the UPLC-HRMS/MS and UPLC-QqQ-MS/MS methods together with the anti-inflammatory evaluation of indolealkylamines. The manuscript is excellent in all its development. The work makes a good contribution to analytical chemistry. The methodology is adequate and the results are solid and supported by experimental work. I would only suggest an extension in the evaluation section of anti-inflammatory activity. Authors should include the data of the positive control used in the experiment (recognized anti-inflammatory compound), and include in the discussion.

Response: Thank you for your suggestion. The data of positive control with brief discussion has been illustrated in the revised manuscript as follows.

“As shown in Figure 5-A, the number of neutrophils in the yolks represents the extent of inflammation. By comparison with the control group (CG), severe neutrophil infiltration was observed in the yolks of the LPS treated group. Generally, a dose-dependent increase in activity was observed in all the test groups. The total IAAs showed obvious anti-inflammatory effects at the dose of 50 μg/mL, compared with the LPS group (P < 0.01). The ratio of anti-inflammatory activity was 35%, it was better than that of positive medicine (indomethacin) with the effect was 25%. When the dose was 100 μg/mL, the activity reached 55% (P < 0.001). Specifically, treatment with N-methyl serotonin displayed a significant ability to decrease the number of neutrophils at all three concentrations (25, 50 and 100 μg/mL), with the activity at 35%, 55% and 60% (P < 0.001), respectively (Figure 5-B and Figure 5-C). However, the bufothionine group showed a relatively weaker anti-inflammatory effect with the activity at 30% in the 100 μg/mL dosage (P < 0.05).

The results aforementioned indicated that the constituents of IAAs did have obvious anti-inflammatory activity, the effect of which even better than positive control group when the concentration over 50 μg/mL. In addition, free IAA N-methyl serotonin showed better activity than combined IAA bufothionine. Given that the content of N-methyl serotonin in total IAAs was much smaller than bufothionine, which led us speculate that free IAAs were the main compounds exert anti-inflammatory activity, probably. Further, we have known that the total content of IAAs, especially free IAAs, in toad venom is much higher than in skin, that’s maybe the reason why toad venom was used for treating infection and inflammation in traditional application. By the way, the anti-inflammatory activity of IAAs was reported for the first time.

Round 2

Reviewer 1 Report

The manuscript could be accepted at the current version.